# *Calluna vulgaris* as a Valuable Source of Bioactive Compounds: Exploring Its Phytochemical Profile, Biological Activities and Apitherapeutic Potential

**DOI:** 10.3390/plants11151993

**Published:** 2022-07-30

**Authors:** Alexandra-Antonia Cucu, Gabriela-Maria Baci, Alexandru-Bogdan Cucu, Ştefan Dezsi, Cristian Lujerdean, Iuliana Cristina Hegeduş, Otilia Bobiş, Adela Ramona Moise, Daniel Severus Dezmirean

**Affiliations:** 1Faculty of Animal Science and Biotechnology, University of Animal Sciences and Veterinary Medicine Cluj Napoca, 400372 Cluj-Napoca, Romania; antonia.cucu@usamvcluj.ro (A.-A.C.); gabriela-maria.baci@usamvcluj.ro (G.-M.B.); cristian.lujerdean@usamvcluj.ro (C.L.); cristina.hegedus@usamvcluj.ro (I.C.H.); adela.moise@usamvcluj.ro (A.R.M.); 2National Institute for Research and Development in Forestry (INCDS) “Marin Drăcea”, 077030 Voluntari, Romania; alexandru.cucu@icas.ro; 3Faculty of Geography, Babeş-Bolyai University, 400084 Cluj-Napoca, Romania

**Keywords:** *Calluna vulgaris*, bioactive compounds, biological activities, therapeutic value, apitherapy, heather honey, antioxidant potential, invasive plant, health benefits

## Abstract

*Calluna vulgaris*, belonging to the *Ericaceae* family, is an invasive plant that has widely spread from Europe all across Asia, North America, Australia and New Zealand. Being able to survive in rigid soil and environmental conditions, it is nowadays considered to be of high nature-conservation value. Known for its nutritional and medicinal properties, *C. vulgaris* stands out for its varied physiochemical composition, spotlighting a wide range of biological activity. Among the most important bioactive compounds identified in *C. vulgaris*, the phenolic components found in different parts of this herbaceous plant are the main source of its diverse pro-health properties (antioxidant, anti-inflammatory, antimicrobial, chemoprotective, etc.). Nonetheless, this plant exhibits an excellent nectariferous potential for social insects such as honeybees; therefore, comparing the bioactive compounds observed in the plant and in the final product of the beehive, namely honey, will help us understand and find new insights into the health benefits provided by the consumption of *C. vulgaris*-related products. Thus, the main interest of this work is to review the nutritional profile, chemical composition and biological activities of the *C. vulgaris* plant and its related honey in order to encourage the future exploration and use of this health-promoting plant in novel foods, pharmacological products and apitherapy.

## 1. Introduction

Being the primary source of food, pharmaceutical products and organic material resources, plants play an outstanding role in the conservation and sustainability of ecosystem services [1]. Throughout these services, plants have a positive influence on the human population and mostly on human wellbeing [2,3,4]. As a consequence, recent decades have witnessed a continuous interest of the scientific field towards medicinal plants, natural products [5,6,7] and apitherapy [8,9] that can provide important resources for sustainable development [10,11]. Because of the importance given to plants for preventing and curing various diseases, their role has grown, and more attention has been given to the secondary metabolites and their nutritional and health benefits [12,13,14,15,16].

In this regard, *Calluna vulgaris* L. Hull (*C. vulgaris*), known also as Common Heather, Ling plant, Heather or Scotch Heather, is utilized for its pharmacological properties, along with other specific characteristics that guarantee its economic value [17]. 

Belonging to the *Ericaceae* family [18,19,20], *C. vulgaris* is native to Europe, but has become one of the most widespread plants all across Asia, North America, Australia and New Zealand [21,22].

Described as an oligotrophic, calcifuge species [23,24], this small evergreen perennial plant with woody, flexible stalks [25] is able to survive in rigid soils (acid soils, poor in minerals) [26,27,28] and severe environmental conditions (precipitations, temperature, altitudinal zonation, etc.) [29,30,31,32].

Measuring a maximum of 100 cm high and having a hemispherical shape, *C. vulgaris* is characterized by multiple-stemmed ascending branches, small and persistent leaves and purple flowers with single, long, inflorescence bunches [33,34] (Figure 1).

The long period of blooming in *C. vulgaris*, namely from May to September–October, makes this invasive plant an excellent source of nectar for pollinators [35], especially bees, that are able to produce around 150–200 kg of honey per hectare [36]. This is a key aspect of the ecosystem services that *C. vulgaris* can provide, taking into consideration the environmental changes faced by both pollinators and plants [37], with critical implications for plant diversity, food and biomass production [38].

As it is a light-dependent plant, its ecological amplitude covers a wide range of areas, including pastures, peatlands, meadows and coniferous woodlands from hilly and mountainous regions [39,40], where the acidic soils can act as a reservoir for water and land-based carbon [41]. However, it exhibits a menacing character for native flora due to the phytotoxicity found in their roots [42] and their capacity to change the habitat and ambient conditions for plant regeneration [43,44].

Through the allelopathic character, namely the interaction of secondary metabolites of plants—of which one is invasive and the others are native [45,46]—perennial plants such as *C. vulgaris* are able to inhibit the germination and growth of native and surrounding plant species, thereby interrupting their regeneration ability [47,48,49]. This process occurs either when different parts of the plant are decomposed or through the rhizome and root exudations, and released into the soil [50]. In addition, the same phytotoxicity is exhibited in *C. vulgaris* tissues and litter that can restrain its own regeneration capacity [51].

Therefore, its invasive character cannot be contested, taking into account that, nowadays, biological invasions can represent a primary menace to biodiversity and nature conservation [52,53,54].

Although it tends to invade large surfaces, its spread is restricted by afforestation and the natural restoration of some native flora [39], or by the negative effects induced through the loss of traditional management [55,56,57,58], agricultural intensification [59], overgrazing, nitrogen deposition [60,61,62] and climate change [63,64].

Still, frequent burning and clipping are considered to be the main alteration factors for the habitats characterized by *C. vulgaris* species and for their related ecosystem services [58,65,66], as it produces a decrease in flower buds [55], a slow vegetative recovery [59], a lower flowering frequency [67] and thereby destroying the heather habitats and their associated fauna. Hence, this plant is nowadays considered to be of high nature-conservation value [68], being a protected habitat in the nature-conservation network of Natura 2000 [69] and assignable to the vulnerable conservation status [70].

Reports in the literature have shown that this plant species is undervalued in terms of its economical, ecological, pharmacological and food potential, as it provides valuable ecosystem services for people and biodiversity that can support sustainable development [71,72,73]. In this respect, Figure 2 points out the key ecosystem services that *C. vulgaris* can provide for both biodiversity and human wellbeing.

Different authors have pointed out that *C. vulgaris* is a habitat for a variety of amphibians, reptiles and insects [75], but also for animals that can be represent a recreation and hunting resource [76,77,78]. Moreover, it represents a source of fodder for grazing animals [76,79], an excellent biomass energy crop [80,81], a methane-emission-reducing species [40,82] and a valuable tourism resource [75,83,84]. 

At the same time, *C. vulgaris* phytotoxicity can act as a competitive advantage in terms of an alternative natural source for the biological control of various weed species [85] or plant diseases [86]. This can be considered an ecofriendly strategy to reduce the use of chemicals in the agriculture sector and thus enhance the crop productivity [87,88]. This kind of bio-herbicide prepared from plants has no damaging effects on crop plants or people’s health [85,87]. For these reasons, the active management of *C. vulgaris* habitats is required in order to maintain its conservation status and its future as a potential mitigator of climate change [40,75,80].

Nevertheless, the potential of invasive plants with regard to their nutritional and bioactive properties has been understudied, in the face of their extensive expansion throughout the entire globe. Thus, the phytochemical composition of *C. vulgaris* has indicated the presence of bioactive compounds such as flavonoids, phenols, tannins, proanthocyanidins, caffeic acid derivatives, steroids, triterpenes, and hydroquinones [22,33,89,90]. These secondary metabolites, which are present in different parts of the plant, vary in composition and pharmacological effect. Hence, positive biological activities have been revealed, including antioxidant [28,89,91,92,93], antimicrobial [28,90,92,93,94,95], anti-inflammatory [95,96,97,98,99,100], neurotropic [101,102] and chemoprotective activities [103], just to mention a few.

Despite the fact that, recently, much of scientists’ focus has been on the chemical constituents and biological properties of *C. vulgaris*, there has not been any systematic review to point out all of the available constituents and the beneficial health properties of this invasive plant, or of its related honey. The prevention and treatment of various disorders using natural products, specifically honey-related products, is known as apitherapy [104,105,106]. This alternative medicinal treatment has been proved to have positive results on different in vivo and in vitro assays, as this overview will point out. Although apitherapy is not yet recognized as a complementary medicine, it represents a promising therapy with a large applicability to the medical field. In this regard, the findings revealed that heather honey exhibits similar properties to Manuka honey [107], and therefore can represent a valuable natural remedy that can be utilized for healing purposes.

In order to enhance the in-depth examination and full use of this plant, the aim of this manuscript is to systematically compile all of the current findings on the *C. vulgaris* plant and honey. Furthermore, the biological activities of the reported plant will be covered to allow us to explore its therapeutic potential, to highlight eventual gaps and to provide a scientific groundwork for future studies that may lead to the creation of functional products based on this plant.

Consequently, to carry out this review, the authors collected data from a wide range of databases such as Web of Science, Science Direct, PubMed, Scopus, and Google Scholar. The selection was unlimited and included up-to-date articles, in order to attain a deeper understanding of the subject.

## 2. Phytochemical Profile of *C. vulgaris*

The *C. vulgaris* chemical composition has been extensively studied in order to identify its main bioactive compounds and nutritive status. Hence, the studies carried out have revealed the presence of different classes of compounds, of which the phenolic compounds are the most noteworthy. Thus, phenolic acids (hydroxycinnamic acid, caffeic acid, coumarins), flavonoids (rutin, quercetin, isoquercetin, kaempferol, luteolin, apigenin), phenols and their glycosides (hydroquinone, arbutin, methylarbutin), tannins, triterpenes, anthocyanidins, terpenoids (lupeol, ursolic, oleanolic acids), organic acids, steroids, and essential oils and their subclasses are the most distinct compounds reported in *C. vulgaris* composition [22,33,90,92,108,109,110,111,112].

### 2.1. C. vulgaris Nutritional Value

*C. vulgaris* represents an integrant part of the pastoral resource throughout Europe and therefore an important grazing source for animals [113,114]. Its nutritive value has long been debated, but only a few studies have reported a complete nutritional profile of *C. vulgaris* [33,92]. As the flowers have been proved to contain the higher nutritive value, Table 1 presents the main macronutrients found in both *C. vulgaris* wild inflorescence and commercial samples.

Thereby, the nutritional profile of wild *C. vulgaris* flowers indicates a great content of fiber (38.96 ± 1.64%) [33], while for the carbohydrates, the commercial samples suggested a higher content (83.1 ± 0.3%) [90] than the wildflower samples (36.21 ± 0.20%) [33]. In general, the proportion of macronutrients in the commercial flower samples is more significant than that in the wild samples; Table 1 points out a considerable difference in protein, fat and ash values. Similar studies have analyzed the nutritional value of various edible plants, suggesting that *C. vulgaris* contains higher amounts of carbohydrates and fiber and thus, greater nutritional value [115,116].

Other important constituents related to the nutritional value of *C. vulgaris* are the fatty acids, which are responsible for the energy intake of the analyzed plant. In this sense, Rodrigues et al., 2018 [33] revealed the presence of various fatty acids in the flowers of *C. vulgaris*, the most predominant being linolenic acid (34.71%), followed by linoleic (27.28%) and palmitic (20.79%) acids, while the less abundant was arachidonic acid (only 1.99%) [33]. These data are in accordance with the results presented by Olechnowicz-Stepien et al. in 1982 [117], but different from what Mandin et al. 2018 [90] suggested, namely that the greater proportion of fatty acids was represented by the polyunsaturated fatty acids (53%), followed by the presence of saturated fatty acids (39.9%), linoleic acid (21%), and palmitic acid (19.6%).

Despite the scarcity of literature regarding the macronutrient profile of *C. vulgaris*, the results shown above emphasize that this evergreen plant can be used as a food ingredient in future products, taking into consideration its high source of energy through the content of fiber and carbohydrates.

### 2.2. Bioactive Profile of C. vulgaris

These bioactive compounds are found in various parts of this perennial invasive plant (roots, flowers, shoots). Worth noting is that the chemical composition of the heather plant is dependent on a series of relevant factors, the most significant being: the climate, season, altitude, soil characteristics or growth stages [28,110,118,119,120,121,122]. In addition, being an invasive plant, *C. vulgaris* can adapt its structure to different environments and consequently, analyzing and comparing samples of the same plant collected from different regions, periods, altitudes or germination stages represents a real challenge.

In this regard, Table 2 summarizes the most relevant research works regarding the bioactive composition of *C. vulgaris*, underlining the territorial localization, the constituents found, the part of the plant analyzed, together with the identification methods and the results obtained, in order to obtain an overview on the phytochemical complexity of the *C. vulgaris* profile.

Jalal et al., 1982 identified the presence of different tannins in both the shoots and roots of *C. vulgaris* [118]. In addition, the same authors observed that the composition and quantity of bioactive compounds may vary according to the season. Thus, in the fresh shoots of *C. vulgaris* was reported a significant variation in the phenolic compounds, with the lowest quantity in January and February—in the form of chlorogenic acid and quercetin glycosides—while from May to December, some other bioactive compounds appeared, specifically (+)-catechin, procyanidin D-l and callunin, and remained constant.

Another study that isolated kaempferol-3-*O*-β-D-galactoside from the aerial part of *C. vulgaris* was reported in Turkey [96]. Similarly, Deliorman-Orhan et al., 2009 [91] found that the major flavonoid present in the ethyl acetate fraction was kaempferol-3-*O*-β-D-galactoside (37.1 ± 0.89%) [91].

Another important group of bioactive compounds is represented by the triterpenoids, which are mostly found in the cuticular waxes of plants [125]. Regarding the *C. vulgaris* plant, Szakiel et al., 2012 [111] reported the presence of these compounds in both the flowers and leaves. The presence of triterpenoid compounds was higher in the leaf cuticular waxes compared to the flower cuticular waxes, with ursolic acid being the predominant triterpene compound of both the analyzed parts of *C. vulgaris*, showing 75.2 ± 4.1 mg/g wax extract in the flower part and 397.7 ± 25.7 mg/g wax extract in the leaf [111]. A representative value for the triterpenoid content is also reported for oleanolic acid (28.2 ± 1.6 mg/g wax extract in the flower, compared to 125.1 ± 9.8 mg/g wax extract in the leaves). Other compounds such as friedelin, lupeol, taraxasterol and taraxerone were also reported in this study, together with some new compounds (betulin, germanicol, 4-epi-friedelin, oleanolic and ursolic aldehydes), which have been isolated for the first time in *C. vulgaris* [111].

Similar results were obtained by Garcia-Risco et al., 2015, who extracted large amounts of triterpenes with supercritical CO_2_, more specifically oleanolic and ursolic acids. From the thirteen extractions, the highest concentration of triterpenic acids was obtained at 30 MPa pressure, 50 °C and at the highest ethanol content utilized (15%), with the highest yield (7.31%) resulting in a content of 141.45 mg/g ursolic acid, followed by 82.87 mg/g of oleanolic acid [123]. These results are higher than those reported previously by Pancost et al., 2002, who found a lower concentration of these two acids (40 mg/g dry matter of ursolic acid and 10 mg/g dry matter of oleanolic acid) [126].

The phytochemical analysis of *C. vulgaris* also showed the presence of vitamers, especially Vitamin E, composed mainly of tocopherols (α-, β-, γ-, δ-tocopherol) and tocotrienols (α-, β-, γ-, δ-tocotrienol) [33]. The importance of this vitamin for human health has been largely discussed; studies have shown its protective characteristics regarding the intracellular defense system and therefore its significance in preventing oxidative-stress-related diseases [127,128]. In this regard, Rodrigues et al., 2018 revealed that the principal vitamer found in the *C. vulgaris* flower extracts was α-tocopherol (32.5 ± 0.48 mg/100 g) [33]. Related reports found that the same vitamin, α-tocopherol, was the most predominant in the dried inflorescences of *C. vulgaris* (5.84 ± 0.07 mg per 100 g dw) [90] and in other invasive plant species [129]. Other important compounds found in the phytochemical analysis of Rodrigues et al., 2018 were quercetin, kaempferol and myricetin derivatives, the most predominant being methoxy myricetin deoxyhexoside, with a value of 308.52 ± 0.48 μg/g [33].

Madim et al., 2018 identified, using different organic and aqueous extracts, various bioactive compounds from the inflorescences of *C. vulgaris* [90]. Thus, of the twelve molecules described, considerable values were reported for myricetin-3-*O*-glucoside and myricetin-*O*-rhamnoside in all of the extracts used, whereas 5-*O*-caffeoylquinic acid had an increased concentration in the methanol, infusion and decoction extracts [90].

Important data on the phytochemical composition of Russian samples of *C. vulgaris* seeds were collected by Cherepanova et al., 2019 [124], who revealed that the main flavonoid present in the analyzed samples was quercetin (30.045 ± 2.003 mg/g), followed by the tannin metabolite catechin (7.679 ± 0.538 mg/g) and hydroxycinnamic acid as chlorogenic acid (4.298 ± 0.301 mg/g) [124].

Starchenko et al., 2020 [102] determined, through the spectrophotometric method, the presence of various phenolic metabolites in the composition of the aerial parts of *C. vulgaris*, of which the dominant ones were arbutin, chlorogenic acid, rutin, (−)-epigallocatechin and (+)-gallocatechin. The content of these compounds was higher in the water extraction (arbutin, rutin) compared with some other metabolites that were more present in the hydroethanolic extracts (70% ethanol), namely chlorogenic acid, rutin, hyperoside, quercetin-3-D-glucoside, (+)-gallocatechin and (−)-epigallocatechin [102] (Table 2). Despite the fact that arbutin was the prevalent compound found in *C. vulgaris* through the water extraction (1.25%), it was less present in the hydroethanolic extract of the plant (70% ethanol) (0.83%), or even undetected [130].

Overall, the presence of phenolic compounds identified in *C. vulgaris* has been reported in all parts of the plant, but their content may vary during different stages of plant growth or with different climatic factors, such as altitude or soil properties.

In this regard, Chepel et al., 2020 [28] monitored the changes of different phenolic compounds found in heather plants, especially in the leaves, stems, roots, rhizomes, flowers, and seeds, at different development stages (vegetative, floral budding, flowering, seed ripening) [28]. The authors pointed out that for the vegetative stage, the highest content of phenolic compounds (especially tannins, flavonoids and proanthocyanidins) was identified in roots (31.66 ± 0.53 mg/g), while the leaves had the lowest quantity of bioactive substances (14.52 ± 1.85 mg/g). Hence, the phenolic content changed with the development of the plant, the leaves gaining more phenolic compounds (flavonoids, anthocyanins, proanthocyanidins and hydroxycinnamic acid) in the floral budding stage (28.15 ± 0.76 mg/g) and the seed ripening stage (32.67 ± 0.12 mg/g), while, as expected, the flowers had the highest total phenolic content during the flowering stage (27.88 ± 0.18 mg/g), for all of the analyzed compounds (tannins, flavonoids, anthocyanins, proanthocyanidins and hydroxycinnamic acid) [28]. These results are sustained by another study on a plant of the same family species, namely *Vaccinium ashei*, which had the highest value of phenolic compounds in the leaves during harvesting time [131]. 

Moreover, because of its wide spread, *C. vulgaris* was analyzed for the altitudinal alteration of its bioactive metabolites [110,119]. Hence, the results of these studies showed that the amount of flavonols increases according to altitude. 

These kinds of studies are very important, because they make a valuable contribution to the understanding of the changes that may occur in the phenolic compound accumulation in various parts of *C. vulgaris* at different growth stages, and therefore promote a better use of this plant.

### 2.3. C. vulgaris as a Bio-Indicator 

It is well known that environmental contamination may have a negative impact on the ecosystem sustainability, and therefore it is very important to analyze the level of trace metals found in different habitats. *C. vulgaris* has been indicated to be sensitive towards metal tolerance and thus has been extensively used as a bio-indicator for the detection of different toxic heavy metals found in the soil or environment, due to the industrial and traffic emissions or the intensification of agriculture by the use of mineral fertilizers [132]. Furthermore, because of the soil characteristics that can ensure the development of *C. vulgaris* plant species, more specifically the high concentration of various constituents that are considered toxic to other species, the survival of native flora can be threatened [48,49]. In this regard, the study of Marks and Bannister (1978) [133] indicated a strong correlation between the constituents found in the soil and their presence in different plant organs of *C. vulgaris*, with lead (Pb) showing its highest concentrations in the woody parts of the analyzed plant. Similar results were published by Matanzas et al., 2021 [134], who investigated the soil and plants from an ancient mercury-mining area in Asturias, Spain and who revealed the ability of *C. vulgaris* to translocate Pb in its leaves [133].

Additional studies have revealed, by using different parts of *C. vulgaris*, the presence of several pollutant elements, including heavy metals. For instance, Mičieta and Murín (1997) [135] proved that in the polluted areas of Slovakia, *C. vulgaris* can be used as a bio-indicator for aluminum (Al), which has the highest observed frequency of unproductiveness of pollen grains in polluted environments (27.4 + 6.1%) [135]. Another important study compared the data from two different periods (1978–1988 and 2011) in the High Tatra mountains of Slovakia [26]. The results showed the presence of different concentrations of elements such as copper (Cu), molybdenum (Mo), zinc (Zn), manganese (Mn), iron (Fe), sulfur (S), fluorine (F), but also chromium (Cr), Pb and cadmium (Cd). These elements showed an increased concentration in 2011 compared to 1978–1988. In both of the analyzed periods, the results reported a positive correlation of Pb, Fe and Cu (r = 0.6320–0.9519) with rising altitude, while Mo, Mn, Cr, Cd and Zn showed no significant correlations with altitude. Regarding S, the highest concentration decreased from >109.8 mg/100 g in 1987–1988 to 87 mg/100 g in 2011 [26]. Similarly, Wojtuń et al., 2017 [136] analyzed the toxicity thresholds for some elements present in vascular plants from Poland, including *C. vulgaris*, revealing that metals such as Cu, Mn and nickel (Ni) were uptaken in the shoots of the plant, with the Mn concentration being the higher compared to the other elements [136]. On the contrary, the results found by Rajsz et al., 2021 [137] in the geothermal areas of Iceland underlined lower mass fractions of Mn (289–659 mg/kg) and titanium (Ti) (129–1180 mg/kg) compared to the results presented by Wojtun et al., 2017 for *C. vulgaris* [136], but higher concentrations of Cr, Cu, Fe and Ni. *C. vulgaris* has also been reported as an arsenic (As)-tolerant species, with the aerial parts of the plant revealing toxic concentrations of this element [138].

Therefore, it can be stated that it is useful to evaluate the presence of trace elements in plants mainly because it can offer an accurate image of the environmental pollution. However, as the acceptable concentrations of emissions of the elements presented above may exceed the normal toxicity amounts, they can represent a threat to human health due to their capacity to infiltrate the human food chain.

## 3. Biological Activities of *C. vulgaris* and Their Therapeutic Potential 

*C. vulgaris* has long been used in traditional medicine due to the therapeutic potential of its phenolic compounds, whose presence is vital in exerting the plant’s biological activity. Furthermore, it is well known that even natural products that are *C. vulgaris*-derived, such as honey, contain high amounts of active compounds that exhibit numerous pharmacological benefits and that are commonly used in alternative medicine, or apitherapy [104,105].

Numerous studies have emphasized these beneficial effects by using extracts from different parts of the plant and testing their applicability in vivo and in vitro study models. Accordingly, as Table 3 points out, the *C. vulgaris* plant has manifested important pharmacological activities such as: antibacterial [28,90,92,93,95], anti-anxiety/antidepressant/neurotropic [101,102], antiviral [123], antiproliferative [139], antinociceptive [140], antioxidant [28,89,91,92,93], antihypertensive and analgesic [99], anti-inflammatory [95,97,98,99,100,140], hypouricemic [99], and chemopreventive and photoprotective activities [97,103,141].

All these valuable properties have been associated with different biologically active compounds and therefore could have promising effects on a wide range of related disorders, as will be presented in what follows.

The **antibacterial** properties of *C. vulgaris* were investigated in disorders related to the urinary tract pathogens where strains of *Escherichia coli* (*E. coli*), *Enterococcus faecalis* (*E. fecalis*) and *Proteus vulgaris* were found to be sensitive to the plant extracts [95]. Similarly, Mandim et al., 2018, 2019 [90,92] highlighted the capacity of *C. vulgaris* inflorescence extracts (acetone, methanol, decoction and infusion) to inhibit twelve pathogenic bacteria strains that are thought to be responsible for urinary tract infections. Among the tested strains, the most susceptible were the Gram-positive bacteria (*E. faecalis*, *Listeria monocytogenes*, methicillin-resistant *Staphylococcus aureus*, *S. aureus*, methicillin-susceptible *Staphylococcus aureus*), which exhibited the lowest minimal inhibitory concentration (MIC) values in the decoction and infusion plant extracts [90,92]. The same authors also investigated the antibacterial activity of the same tested plant extracts in vaginal microbial strains (*Gardnerella vaginalis* and *Neisseria gonorrhoeae*) illustrating that the MIC value for the *Lactobacillus* strains was higher (20 mg/mL) than the values for the two tested vaginal microbial strains (*Gardnerella vaginalis* had a value of 5 mg/mL and *Neisseria gonorrhoeae* had a value of 2.5 mg/mL) [90,92]. This indicates that *C. vulgaris* can inhibit the tested pathogens without affecting the normal balance of vaginal microbiota. More recent studies have proved that the antibacterial capacity of *C. vulgaris* may vary according to the plant organ from which the extract was taken and the growth phases. In this regard, Chepel et al., 2020 [28] suggested that both the Gram-negative bacteria *E.coli* and the Gram-positive bacteria *Bacillus subtilis* (*B. subtilis*) are more susceptible to leaf and stem extracts (during vegetative and floral budding stages), while the root and flower extracts manifested a weak antibacterial potential against the two tested bacteria (at the stages of floral budding, flowering and seed ripening against *B. subtilis* and at the stage of seed ripening against *E. coli*) [28]. The inhibition capacity of *C. vulgaris* against human pathogen microorganisms was also revealed by Varga et al., 2021 [93], who proved that different extracts of the plant could inhibit the growth of two two Gram-positive and three Gram-negative strains [93]. Thus, the chloroform, ethyl acetate, butanol and water extracts were more effective against *S. aureus* and methicillin-resistant *S. aureus*, the butanol and water fractions inhibited *E. coli* strains, while *E. coli ESBL* and *Klebsiella pneumoniae ESBL* were more sensitive to the water extract [93].

The depression-associated disorders were studied by Saaby et al., 2009 [101], who found out that the main biologically active compound responsible for the **nerve-calming** effects of the heather plant was quercetin, which was able to inhibit monoamine oxidase A (MAO-A) in the ethyl acetate phase (the half-maximal inhibitory concentration of quercetin for MAO-A was 18 ± 0.2 μM) [101]. It is worth mentioning that MAO-A is the main compound responsible for the alteration of important neurotransmitters, such as serotonin, dopamine and norepinephrine [142], which are vital for nerve cells and normal brain functioning, thus the inhibition of MAO-A may have crucial effects in preventing depression [143,144]. Similar antidepressant, anti-anxiety, neurotropic and anxiolytic effects were observed by Starchenko et al., 2020 [102] using an in vivo study model on rats and mice. The hydroethanolic dry extracts of *C. vulgaris* (ethanol 70%) were administrated to the above-mentioned study models before testing their behavior using various compartmental tests [102]. Thus, the **anti-anxiety** effect was demonstrated by the fact that the administered dry extract of *C. vulgaris* determined the orienting–research behavior of the tested animals, while the antidepressant effect was investigated by the animals’ capacity to increase the time of active swimming and reduce the immobility time [102]. Moreover, the **anxiolytic** effect of the plant extracts was determined by analyzing the number of approaches of the animals to the cube and the time spent on cube examination. All of these neurotropic effects are strongly correlated with the presence of phenolic compounds such as: arbutin, chlorogenic acid, rutin, hyperoside, quercetin-3-D-glucoside, (+)-gallocatechin and (−)-epigallocatechin in the composition of the used extracts of *C. vulgaris* [102].

Other pharmacological effects of *C. vulgaris* include **antiviral** properties indicated by the presence of oleanolic and ursolic acid, which showed a high Hepatitis C virus inhibition in the ethyl acetate fractions [123]. In addition, the same ursolic acid isolated from heather flowers also showed **antiproliferative** activity, as assessed by Simon et al., 1992 [139]. The authors revealed that this compound has an increased role in inhibiting the lipoxygenase activity and thus the proliferation of HL60 leukemic cells in a dose-dependent manner.

Worth mentioning is the **antioxidant** capacity of *C. vulgaris*, as several studies underlined that the phenolic compounds are the main direct links to this property. For instance, Deliorman-Orhan et al., 2009 [91] tested the scavenging activity of the kaempferol-3-*O*-β-D-galactoside flavonoid against some reactive oxygen radical species, showing 90.1% and 93.5% inhibition of 1,1-diphenyl-2-pirylhydrazyl (DPPH) radicals at 200 and 1000 μg/mL, respectively, while the inhibition of O^−2^ was lower—70.4% and 79.8%—using the same concentrations of the flavonoid [91]. On the other hand, the research of Dróżdż et al., 2016 [89] showed that the antioxidant capacity of the wild heather plant is lower than the cultivated one, with the total reducing capacity of *C. vulgaris* extracts ranging from 75.7 mg GA/g (for wild plants) to 89.1 mg GA/g for cultivated plants [89].

As studies have revealed, the antioxidant activity is correlated to the content of the phytocomponents present in the analyzed samples. Thus, as the research of Chepel et al., 2020 [28] showed, the highest degree of correlation for the antioxidant activity using the FRAP (ferric-reducing antioxidant power) assay was exhibited for total phenolic content (r = 0.77, *p* ≤ 0.01), while for the DPPH assay, the correlation for the phenolic compounds was (r = 0.65, *p* ≤ 0.01) [28]. Similar results were published by Varga et al., 2021, who demonstrated the same positive correlation between the antioxidant activity and the bioactive compounds [93].

The **antigout** potential of *C. vulgaris* is manifested through the antihypertensive, analgesic, hypouricemic and anti-inflammatory effects, as Vostinaru et al., 2018 [99] evaluated in their study. The authors tested these effects in vivo, using an ethanolic extract from the *C. vularis* plant that was applied to different rat model experiments. Accordingly, the ethanolic extract of the plant, of which the major component was chlorogenic acid, confirmed the above-mentioned effects at the dose of 500 mg/kg plant extract [99].

As for the **anti-inflammatory** potential, *C. vulgaris* showed beneficial effects on various disorders such as gastritis, ulcers [140], urinary tract pathogens [95], and Alzheimer disease [98]. For instance, Orhan et al., 2007 indicated that kaempferol-3-*O*--d-galactoside was the main bioactive compound related to the anti-inflammatory effect of *C. vulgaris* in a mice experiment model [140]. Instead, Ghareeb et al., 2014 validated that *C. vulgaris* has the ability to inhibit acetylcholinesterase and thus could be integrated into the treatment of dementia in patients with Alzheimer’s disease [98].

Likewise, the **chemoprotective** effect of the reviewed plant was evidenced in different studies that emphasized that the *C. vulgaris* extract can be used for skin burns and skin cancer prevention, as part of a hydrogel before UVB exposure [141], or as an inhibitor agent of UVB-induced cell death [97]. Similarly, Virag et al., 2015 [103] underlined the **photoprotective** effects of heather by suggesting that it acts as a suppressive to the production of reactive oxygen species because of its compounds, namely hyperoside, quercitrin, quercetin and kaempferol [103].

As highlighted above, the increasing interest in plants, and especially *C. vulgaris*, as natural sources with pharmacological effects is manifested in a plethora of studies that confirm the valuable biological effects of this plant. All of these studies aim to provide new data for the potential medicinal use of *C. vulgaris* in the treatment of numerous human disorders.

## 4. *C. vulgaris*—From Plant to Honey

Because of their important constituents, plants, in general, represent an important source of food for various insects. Honeybees in particular are capable of producing numerous products that are beneficial to human health [145], the most important such product being honey [146].

As was described in the previous sections, the *C. vulgaris* plant contains a large spectrum of secondary metabolites that are present in all of the analyzed parts of the plant. When collecting the nectar from the flowers, the bees transfer the bioactive substances present in their nectary glands to the honey, together with all of the specific compounds contained by the plant [147,148], creating an outstanding product with countless benefits for human health.

*C. vulgaris* is considered to be of major importance for apiarists, both because of the long blooming period and the large quantity of nectar production [35,149,150]. Therefore, bees can produce substantial honey quantities, compared to other nectariferous plants [36].

Moreover, the structure of the *C. vulgaris* plant has anthers that contain breaches that make possible the extraction of pollen and nectar by various insects, especially honeybees [151]. Consequently, the rich source of pollen and nectar contained by this plant could explain the foraging behavior of honeybees [152,153].

It is well known that honey composition is highly related to and influenced by the botanical and geographical origin of plants [154,155,156], which has a great impact on some characteristics such as flavor, texture and color [157,158]. Therefore, comparing the bioactive compounds observed in the heather plant and in its related honey is of major importance due to the attention that natural products have gained lately, and the unlimited opportunities for their use as therapeutic agents. In addition, the clinical observations that will be pointed out in this chapter have a considerable contribution to future exploration studies in which the *C. vulgaris* plant and honey can be part of health-promoting therapies such as apitherapy.

### 4.1. Heather Honey and Its Physical–Chemical Parameters

Being a “natural sweet substance produced by *Apis mellifera* bees” [159], especially from the nectar of plants, blossom honeys such as heather honey contain various constituents (sugars, enzymes, amino acids, organic acids, carotenoids, minerals, vitamins, volatile compounds, flavonoids and phenolic acids) that are recognized for their nutritional and medicinal profiles [160,161,162]. Moreover, because sometimes bees collect most of the nectar from a specific plant, the honey produced is called monofloral honey and it acquires the same individual characteristics as the main plant [163,164]. Some of these characteristics are related to the color, fragrance or aroma, in addition to the pharmacological properties, which are attributes that are considered to be the main motivation factors of consumer choice [165,166,167,168,169,170,171].

Therefore, monofloral honeys such as heather honey have witnessed an increased demand over the last decades, expanding their market value [171,172,173,174,175], but at the same time giving rise to fraudulent mislabeling [156,176,177].

Besides the chemical parameters, the honey structure may be influenced by other external conditions such as the climate, environment or beekeeper knowledge about the good practices that should be maintained in order to keep the honey quality unspoiled [178]. Therefore, various methods were developed to prevent honey discrimination, mostly by analyzing the floral and the geographic origin [9,155,158,179,180,181,182]. These methods are extremely important for determining the quality and authenticity indicators of honey, especially when honey is used for therapeutic purposes.

Among all the monofloral honey varieties, heather honey has atypical and unique physical–chemical properties. One of the most distinctive characteristics of this type of honey is called thixotropy [25,150,183,184,185,186]. This physical property is a consequence of the high content of colloidal proteins [187,188] that makes it very difficult to extract. As a consequence, its consistency changes when mechanically stressed, whereas it has a gelatinous appearance in the combs [189]. The extraction process is performed with the help of a special instrument (Figure 3), similar to a brush that penetrates the honey combs cells and enhances the fluidity of the honey, allowing it to flow [150]. One of the reasons for its higher price compared to the other honey varieties present on the market is the challenge in extracting this type of honey [190].

When analyzing the physical–chemical parameters of heather honey, regardless of the geographical origin, it can be stated that it is a distinctive type of honey, with unique features that are mostly related to: the electrical conductivity, free acidity, color, moisture content, free sugar composition, and 5-hydroxymethylfurfural (HMF) content [25,191,192], as described in Table 4.

The EC of honey is strongly related to its botanical origin and therefore represents an important quality control for honey authentication [199]. This parameter is influenced by the concentration of organic acids, minerals, ash content and proteins present in the honey’s composition [199,200,201], and its proportion depends on the plant’s geographical origin [202]. In addition, both the ripening and storage time of honey can influence the EC quantity [203]. In general, heather honey possesses one of the highest ECs compared to the other honey varieties, with the only exception being chestnut honey, which surpasses all nectar honeys in terms of this feature [189,192]. The highest electrical conductivity has been observed in Spanish heather honey by Escuredo et al., 2019 [196] with a value of 0.777 ± 0.11 mS/cm, compared to the lowest identified in Estonia (0.4 ± 0.2 mS/cm). The Council Directive 2001/110/EC [159] related to honey specifies that the electrical conductivity of heather honey can be situated beyond the 0.8 mS/cm limit, as encountered in the above-mentioned studies in Table 4.

Mineral content can also be an essential feature in the determination of the electrical conductivity and thus of the botanical origin of honeys [204]. For instance, Nozal et al., 2005 [205] reported that it can be differentiated between several honey varieties such as: heather, rosemary, thyme, lavender, and honeydew honeys by establishing the amount of certain minerals, namely magnesium (Mg), calcium (Ca), aluminium (Al), iron (Fe), manganese (Mn), zinc (Zn), boron (B), copper (Cu), cobalt (Co), chromium (Cr), nickel (Ni), cadmium (Cd) and phosphorus (P) [205]. Similarly, Dezmirean et al., 2010 [25] revealed that the highest mineral content present in Romanian *C. vulgaris* honey was potassium (K). These results are similar to those obtained by Moise et al., 2013 [193] for the same Romanian honey variety, but from different geographical areas, as well as to those of Ghorab et al., 2021 [197]. These outcomes underline the fact that heather honey represents an excellent source of minerals for everyone that consumes it. In addition, vitamins are also present in the chemical composition of heather honey, particularly vitamin C, which has been proved to be the most predominant in Portuguese honey samples [191,206].

The free acidity is associated with the content of organic acids present in the honey samples, with gluconic acid being the major organic acid found in this honey variety [25]. Table 4 reports a wide range of acidity values for the tested heather honey samples and as it may be seen, all the samples contained levels of free acidity below the level imposed by the European regulations, namely 50 mL of acid equivalent per 1000 g [207]. The observed values ranged from 4.40 ± 0.03% free acidity in Turkey [194] to 34.7 ± 10.2% free acidity for the Algerian heather honey samples [197]. These data indicate that the tested honey samples do not undergo any fermentation process.

Regarding the honey color, it is associated with the content of the mineral elements, phenolic compounds and pollen/nectar botanical origin [165,198]. Heather honey has an interesting color, similar to maple syrup, graded from a light to dark amber color [188]. On the other hand, the smell is quite strong and the flavor is bittersweet [189,208,209]. As described in Table 4, the darkest heather honey samples were identified in Portugal (333 ± 0.00 mm Pfund) [191], while the lightest samples were from Poland (69 mm Pfund) [192].

The moisture content in honey’s composition is considered to be one of the most significant criteria for its quality, as it can affect the storage conditions and thus determine honey fermentation [210,211,212]. Despite the limits established by the European regulations regarding the maximum moisture limit allowed for commercial honeys—20% (g/100g)—[159], in heather honey this parameter is exceeded, reaching up to 23% of its composition [150]. A similar exceedance was reported in Italy for citrus honey (20.56%) [213]. The studies presented in Table 4 do not indicate a moisture level exceeding 21%, except the average ± standard deviation values of 24.20 ± 0.00% identified in Portugal [196] and 23.33 ± 1.58% in Ireland [188]. Besides this, Dezmirean et al., 2010 [25] observed in the Romanian samplings an average moisture value of 20.66 ± 1.02%, while the lowest value was found in Portugal 16.47 ± 0.06% [191].

As for the carbohydrates, honey contains sugars that represent 95% of the dry matter [212], especially in the form of fructose and glucose [147], which ensures honey’s nutritive value. The concentration of sugars present in honey varieties depends on the botanical origin of the flowers and it can be negatively influenced by the conditions and the time of storage [148]. Hence, the sugar content in honey is one of the most important elements in identifying honey quality. 

Regarding heather honey composition, it reveals a high content of fructose over glucose [189,208]. This proportion between fructose and glucose is usually employed in the classification of monofloral honeys [214]. In the above-mentioned studies, the amount of total sugar was relatively similar for all the analyzed heather honey samples, with a slight difference in favor of Portugal, which had a Brix value of 82.03 ± 0.06% [191]. The exception was for Spain [196] and Algeria [197], which did not measure this parameter. However, all the samples met the European quality requirements (above 60 g/100 g).

The last analyzed feature is represented by the 5-hydroxymethylfurfural (HMF) content that is produced during the acid-catalyzed dehydration of sugars [195]. This chemical reaction is due to the honey temperature [215] and storage conditions [162,216].

The 5-HMF reaction is considered to be strongly related to the higher free acidity values [189]. 5-HMF is a parameter that establishes the honey freshness [198,216,217,218,219] or the possible adulteration of honey with inverted sugars [162], its content being established by the European legislation, namely the *Codex Alimentarius* Standard Commission (5-HMF in honey at 40 mg/kg and 80 mg/kg for tropical origin honeys) [220].

Among the samples listed in Table 4, the only sample that exceeded these limits was the heather honey from Turkey, which had a 5-HMF content of 94.66 ± 1.36 mg/kg [194], which can be explained by the fact that the honey was heated or inappropriately stored.

### 4.2. Bioactive Polyphenols and Volatile Compounds Specific to Heather Honey 

#### 4.2.1. Polyphenols

The biologically active compounds found in honey are the substances that are related to the beneficial properties exhibited on human health, with the antioxidant, anti-inflammatory and antibacterial effects [221,222,223] being the most common. The majority of these substances are transferred from the plant into the honey through the nectar and thus their composition is correlated with the geographical and botanical origin of the plant source [212,224,225,226]. Nevertheless, the composition and amount of these bioactive substances can be affected by other external factors such as the season, temperature, storage conditions, harvesting period, etc. [148,212].

Depending on their chemical structure, these biologically active compounds can be classified into phenolic acids and flavonoids [224,227], that in turn have other sub-divisions and different chemical structures [214,228].

Regarding the specific biologically active compounds found in heather honey and that can be considered as authentication tool markers, studies have revealed the presence of cis/trans-abscisic acids in samples of Portuguese heather honey [229,230]. Accordingly, Ferreres et al., 1996 pointed out the presence of various flavonoids, finding that myricetin, myricetin-3-methylether, myricetin- 30-methylether, and tricetin are the uppermost compounds that are related to the heather honey floral origin [229,230]. The results are similar to those described by Jasicka-Misiak et al., 2012 [231], who found, besides abscisic acid, many more phenolic compounds in two Polish monofloral honeys, which can be considered potential markers for the heather honey. In this regard, the most worth mentioning are: 3-hydroxybenzoic acid, 4-hydroxybenzoic acid, chlorogenic acid, syringic acid, vanillic acid, ellagic acid, rosmarinic acid and myricetin.

Furthermore, Can et al., 2015 [217] reported notable values in Turkish heather honey for p-coumaric acid (2.98 ± 1.60 µg/100 g) and quercitin (21.05 ± 0.512 µg/100 g), while, compared to the Polish heather honey samples, apigening, chlorogenic, vanillic and ferulic acids were not detected [231]. On the contrary, Halagarda et al., 2020 [232] identified higher values for p-coumaric acid (407 ± 8 µg/100 g), cystin (108 ± 7 µg/100 g), galangin (154 ± 34 µg/100 g), apigening (45 ± 1 µg/100 g) and quercetin (31 ± 3 µg/100 g) in the samples of heather honey from Poland. Similar results have been reported for Latvian heather honey in terms of the levels of p-coumaric acid (2519 ± 738 μg/kg) and quercetin (198 ± 86 μg/kg) [233]. Even so, there were dissimilarities in the other phenolic compounds such as luteolin (higher values in Latvian samples), chrysin, galangin, and apigenin (higher values in Polish heather honey samples).

Rodrigues da Silva et al., 2021 [234] identified 17 phenolic compounds in different Portuguese honey heather samples; the hydroxybenzoic acids were the most predominant phenolic acids, while the catechin derivative was the main flavonoid present in the analyzed honey.

Another phenolic compound that can be considered a marker for heather honey is 3-hydroxyphenylacetic acid, which was identified in the study of Vazquez et al., 2021 [226]. This compound had a value ranging from 54 to 242 μg/g for Galician heather honey, being the most predominant phenolic compound in all of the 91 analyzed samples, and thus a specific authenticity marker for this honey variety.

As it can be seen, there are still differences between the compositions of bioactive compounds, differences that may be explained by the influence of the geographical origin of the plant source and thus of the analyzed honey samples [216].

#### 4.2.2. Volatile Compounds

The volatile compounds present in different honey varieties are aromatic substances that are acquired from the nectar obtained by the bees from the foraged flower [158,235]. However, their composition can vary depending on some chemical reactions that take place during the honey thermal processing, extraction or storage period [212,236]. These aromatic substances can include fragrances, aromas and pharmacological properties, which may have an important impact on consumer choice [237]. In addition, these substances can be used as botanical source indicators [212,238].

The most important volatile aromatic compounds that were observed in various honey varieties include: aldehydes, acids, ketones, esters, alcohols, linear hydrocarbons, and cyclic compounds [182]. In the case of heather honey, studies have described the presence of different volatile compounds that could be correlated with the botanical origin of *C. vulgaris*. For instance, guaiacol, p-cresol, propylanisole and p-anisaldehyde were reported for various Spanish monofloral honey varieties, including citrus, rosemary, eucalyptus, lavender, thyme and heather honeys [239], while isophorone and 2-methylbutyric acid were reported only for heather honey [209]. 

Another finding was reported by Plutowska et al., 2011, where the presence of several volatile compounds, namely: 3,4,5-trimethylphenol, phenylacetic acid, benzoic acid, b-damascenone and isophorone (3,5,5-trimethyl-2-cyclohexen-1-one) were observed in the chemical composition of heather honey from Poland [240].

Moreover, Karabagias et al., 2018 indicated that the total volatile compounds found in Portuguese tested honeys were mostly assigned to the heather honey variety, of which the most predominant compounds were heptane, benzaldehyde, benzeneacetaldehyde, cis-linalool oxide, and hotrienol [191].

### 4.3. Biological Activities of Heather Honey and Its Potential Use in Apitherapy

As studies have revealed, the biological activities of honey are in general associated with the presence of phenolic acids and flavonoids, which exhibit a great influence on the wellbeing of humankind [162,170]. The most notable such beneficial properties are the antioxidant, anti-inflammatory and antimicrobial effects, which are commonly used in alternative medicine, including apitherapy [208].

The use of bee products as therapeutic agents, known as apitherapy, is not a recent practice, as natural products and mostly honey have been used for preventing or treating various disorders or for beauty benefits [241]. The therapeutic benefits of honey have been employed by ancient civilizations whose knowledge has been widespread for thousand years up until present times, when the compositions of diverse honey varieties are being widely investigated for their medical potential [217,218,223,242,243].

Heather honey, like its source plant *C. vulgaris*, possesses plenty of biologically active compounds that are responsible for the pharmacological activities that this type of honey exhibits, as will be presented below.

#### 4.3.1. Antioxidant Activity

The antioxidant activity of honey depends on the composition of different classes of secondary metabolites that prevent or inhibit various oxidative mechanisms in living organisms [244]. Such metabolites are the phenolic compounds, whose antioxidant capacity is related to the plant species, harvest season, climatic conditions and storage [131,216,217,218]. As will be presented in the following lines, the antioxidant activity varies between the geographical areas of the analyzed honey samples. This fact suggests that the geographical origin with different soil and climatic parameters has a huge influence on the antioxidant property of honey in general, and particularly of heather honey. 

Concerning the evaluation of the antioxidant activity of heather honey, the most employed assays are the ferric-reducing antioxidant capacity (FRAP), the radical-scavenging activity (DPPH) and the radical-cation-based assays (ABTS). 

For instance, Wilczyńska, 2010 [245] evaluated the antioxidant capacity of thirty-two Polish honey samples, of which the three samples of heather honey showed the highest DPPH-radical-scavenging activity (100%), while in the ABTS test, this property varied from 21.42 to 31.51%. In this study, the correlation between the antioxidant capacity of heather honey and the phenolic content was a positive one (r = 0.74 for TPC/DPPH, r = 0.55 for TPC/ABTS^•+^, *p* = 0.05), proving once again that the phenolic compounds are the substances that most influence this property [245]. The radical-scavenging activity measured in terms of DPPH for heather honey presented in this study is higher than the values obtained by Moise et al., 2013 for the Romanian heather honey samples, ranging from 49 to 61% [193], and for the Portuguese samples (varying from 33.2 ± 0.1% in 2012 to 31.0 ± 1.3% in 2016) [246].

Other Polish heather honey samples were analyzed by Kuś et al., 2014 in comparison with other five honey varieties such as black locust, rapeseed, lime, goldenrod and buckwheat, revealing that, besides buckwheat, heather honey exhibited the highest antioxidant potential in both the DPPH (0.6 ± 0.1 mmol TEAC/kg) and FRAP assays (2.1 ± 0.5 mmol Fe2+/kg) [247]. Both the content of polyphenols and the color parameters were significantly correlated with the antioxidant activity of heather honey, as studies have indicated that the darker the color of the honey, the more polyphenols it contains [248]. Moreover, the studies on Turkish monofloral honeys, including heather, evidenced the same existing correlation between the phenolic content and the antioxidant activity of various honey samples [216]. In this case study, the heather antioxidant capacity was exceeded by chestnut and oak honeys in terms of FRAP values, while for the DPPH assay, heather honey had one of the lowest values compared with the other thirteen analyzed honey assortments [216]. On the other hand, from the Portuguese eucalyptus, chestnut and heather honey samples, heather had the highest phenolic content and thus the highest antioxidant property, measured in terms of DPPH-scavenging activity (83.75 ± 0.01%) [191].

Comparatively, the study performed in the Galicia region of Spain on six different honey types showed a higher antioxidant capacity for honeydew (420–1017) and chestnut honeys (140–54) expressed as micromoles of Trolox equivalents (TRE) per 100 g of honey, while heather honey contained 132–392 μmol TRE/100g [226].

Another assay employed by Starowicz et al., 2021 for the evaluation of the antioxidant potential of heather honey was the photochemiluminescence technique (ACW) [249]. In this study, the highest value for the superoxide-anion-radical-scavenging capacity in the ACW system was reported for buckwheat (289.8 ± 14.1 μmol Trolox/g) and heather (119.9 ± 2.51 μmol Trolox/g) honeys, while the lowest value was identified in acacia honey sample. As for the FRAP assay, it provided an average value of 47.13 μmol Trolox/g for buckwheat honey and 31.58 μmol Trolox/g for heather honey, in contrast to rapeseed honey, which had the lowest identified FRAP value [249]. These findings are in agreement with the study of Kivima et al., 2021 that found an average level value of water-soluble antioxidants (ACW), for the Estonian tested honey samples, of 115.2 ± 67.9 mg AAE/100g. These samples contained substances that were specific to honeydew (cinnamic acid) and heather honey (myricetin), thus the correlation with these two honey varieties. In addition, the same study showed that heather honey was associated with the lowest lightness parameter, which suggested once again its darker color and thus the interdependence with the antioxidant property [198].

One interesting study was performed by Rodrigues da Silva et al., 2021, who tested the antioxidant capacity of *C. vulgaris* honey from Portugal and its effect on the oxidative stress manifested in human erythrocytes. Thus, the study showed that heather honey could scavenge the reactive oxygen species (ROS) and reactive nitrogen species (RNS) through different assays such as: superoxide radical (O2^−^), hydrogen peroxide (H_2_O_2_), hypochlorous acid (HOCl), singlet oxygen (^1^O_2_), nitric oxide (NO) and peroxynitrite anion (ONOO^−^). From all of these assays, the highest antioxidant capacity of heather honey in the ROS was 95 ± 2 μg/mL (IC_50_) for the HOCl assay, whereas for the RNS the highest scavenging activity was against the ONOO^−^ assay with a value ranging between 29 ± 2 and 32 ± 1 μg/mL. Moreover, the authors proved that the *C. vulgaris* honey extract was able to inhibit hemoglobin oxidation (158 ± 2 μg/mL), lipid peroxidation (36 ± 1 μg/mL) and hemolysis (19 ± 1), in a dose-dependent manner (IC_50_) [217].

All of these mentioned studies and results demonstrate that heather honey has a strong antioxidant potential that can be further evaluated in vitro and in vivo model studies, which will have a great impact on the medical field, especially in treating oxidative-stress-associated disorders.

#### 4.3.2. Antibacterial Activity

Another important biological property of honey that exhibits valuable therapeutic effects is the antibacterial capacity. This property is mainly associated with the geographical and botanical source of plants, bee species or the processes that honey goes through before it arrives at the customer [250]. Some of the most important factors that have a significant contribution to the antibacterial activity of honey are: the high osmolality and sugar content, low pH, viscosity, water activity, polyphenols and methylglyoxal (MGO) content, hydrogen peroxide activity, and the bee immune-enhancing peptide [251,252,253,254]. All of these features can influence honey composition and the synergic action of its compounds. Moreover, because antibiotics can no longer inhibit all of the bacterial strains [255], natural products such as honey have regained the interest of the research community.

In this direction, heather honey has been proved to have remarkable antimicrobial capacity. Serving as an example, Feás et al., 2013 tested the inhibition activity of Portuguese heather honey against four bacteria: *Bacillus cereus* (*B. cereus*), *S. aureus*, *E. coli*, and *Pseudomonas aeruginosa* (*P. aeruginosa*), and four yeast species: *Candida famata* (*C. famata*), *Candida neoformans* (*C. neoformans*), *Candida krusei* (*C. krusei*), and *Candida albicans* (*C. albicans*). The results showed that heather honey manifested growth-inhibition effects for all of the tested strains, especially for the Gram-positive bacteria (*B. cereus*—MIC = 1.51 ± 0.90%; *S. aureus*—MIC = 4.73 ± 2.09%). As for the yeast species, the most resistant to the honey effect were *C. albicans* (MIC = 23.33 ± 10.31%) and *C. famata* (MIC = 22.38 ± 9.14%), while *C. krusei* was the most sensitive (MIC = 14.33 ± 5.68%) [256].

In addition, comparable results were published by Dezmirean et al., 2015, who showed that heather honey samples from Romania had the highest inhibition effects (20μL honey solution, 80%) for the same Gram-positive bacteria *B. cereus* (MIC average = 2.1%). On the contrary, *Salmonella typhi* had the highest MIC (7.4%). The same authors also measured the inhibition diameter of heather honey against the tested microorganism, revealing that Gram-negative bacteria such as *Salmonella enteritidis* had the highest diameter (11 mm), while *Salmonella typhi* had the lower inhibition diameter zone (9.3 mm) [252].

A most recent comparative study between the antimicrobial and antifungal effects of Manuka and Portuguese heather honey varieties against single biofilms of *Candida tropicalis* (*C. tropicalis*) and mixed biofilms with *P. aeruginosa* was employed by Fernandes et al., 2020 [257]. These effects were tested using different honey concentrations. Thus, a cell reduction of the two tested bacteria was obtained with a honey concentration above 25% (*w*/*v*), with heather honey showing a higher antifungal effect for single-species biofilms of *C. tropicalis*, while Manuka honey also exhibited this activity in the mixed biofilms with *P. aeruginosa*. As for the antimicrobial effect, Manuka honey again had the highest inhibition for both single and mixed biofilms at a concentration of honey of 50% (*w*/*v*). However, the combination of honey with antifungals and antibiotics showed no inhibition result [257].

Going further, Shirlaw et al., 2020 tried to evaluate the antibiofilm and antivirulence potential of some constituents of the above-mentioned types of honey, namely Manuka and heather, against several Gram-positive and Gram-negative species biofilms, including *P. aeruginosa*. The results confirmed that at a minimal inhibition concentration (0.25 mg/mL), both the Manuka and heather honey types inhibited the biofilm formation in the majority of the tested strains, except for the *S. aureus* strain, which increased with the Manuka honey, and the *E. faecalis* bacteria, which was stimulated by the heather honey. However, both types of honey inhibited the biofilm formation in *P. aeruginosa* mainly because of the presence of oleanolic acid (associated with heather honey), benzoic acid (present in both honeys) and MGO (characteristic of Manuka honey) [107].

Overall, the antimicrobial activity of honey, and particularly of heather honey, demonstrates that bee products could play a key role in alternative therapies. Therefore, understanding the factors and mechanisms involved in the antimicrobial activity of honey is essential for the development of new products and biomaterials that could enhance its health-promoting applications. Moreover, it suggests that heather honey could be employed as an antifungal agent, especially in skin disorders, keeping in mind that in order to assess its contribution to the medical field (particularly apitherapy), future clinical studies on the impact that heather honey has on treating different disorders are still needed.

## 5. Conclusions and Future Perspectives

The scientific interest regarding *C. vulgaris* plant has emphasized the large spectrum of ecosystem services that it can provide, in terms of sustainable economic and ecological development. Moreover, this review reported the nutritional and phytochemical profile of *C. vulgaris* and particularly the phenolic components found in all of the parts of this herbaceous plant, trying to form a connection between the variety of pharmacological activities and its main bioactive compounds. The promising pro-health benefits of the *C. vulgaris* plant were also pointed out, focusing on its antibacterial, anti-anxiety, anti-inflammatory, antioxidant and chemoprotective effects, just to name the most important ones.

As this plant exhibits excellent nectariferous potential for honeybees because of the long blooming period, our review brings forth a natural, functional and undervalued product, namely heather honey. Thus, we also evaluated the specific bioactive compounds of heather honey and their related medicinal properties, in order to demonstrate once again the high quality of this sort of honey.

Despite the promising information on the chemical and pharmacological composition of heather plant, the studies regarding its related honey as well as its potential beneficial activities are still scarce. That is why we consider that this review may contribute to a starting point for future studies regarding the health benefits provided by the consumption of *C. vulgaris*-related products, notably heather honey, and for the introduction of this natural product into novel pharmacological products. In this regard, evidence-based pre-clinical and clinical trials are necessary in order to explore the potential applicability of heather honey for medical use and apitherapy. To the best of our knowledge, this is the first review that brings together *C. vulgaris* plant and honey and their therapeutic potential.

## Figures and Tables

**Figure 1 plants-11-01993-f001:**
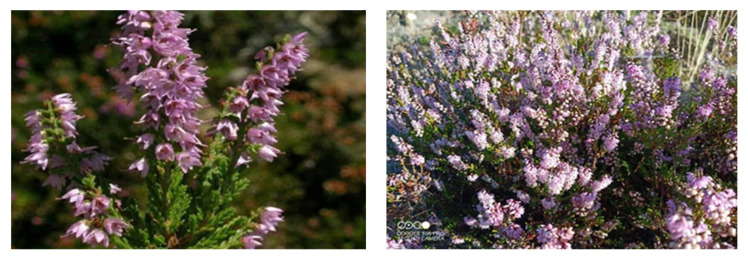
*C. vulgaris* plant (Mihai Grama, beekeeper—personal collection).

**Figure 2 plants-11-01993-f002:**
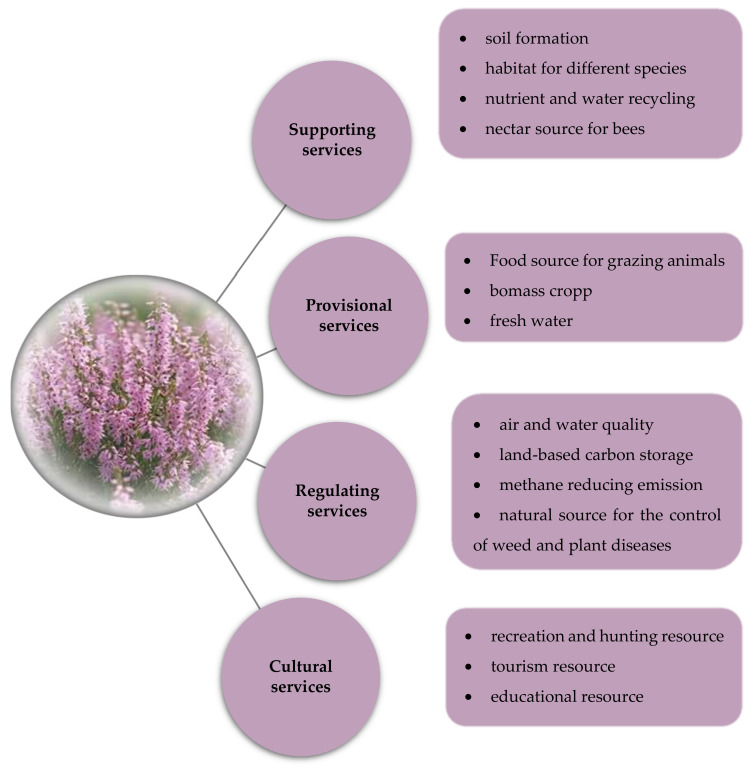
Ecosystem services provided by *C. vulgaris* (adapted from Bonn, 2010) [74].

**Figure 3 plants-11-01993-f003:**
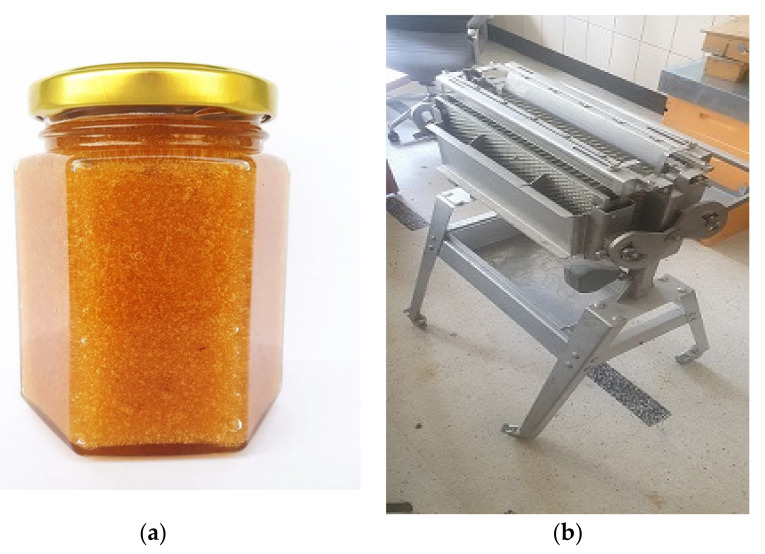
(**a**) Heather honey (Săbăduş Ioan, beekeepr—personal collection), and (**b**) heather honey extractor (Beekeeping and Sericulture Department, Faculty of Animal Science and Biotechnology, UASVM Cluj-Napoca, Romania).

**Table 1 plants-11-01993-t001:** Nutritional composition of C. vulgaris flowers: comparison between wild and commercial samples.

Wild Samples [33]		Commercial Samples [90]	
Nutritional Value (g/100 g)		Nutritional Value (mg per 100 g dw **)	
Ash	2.31 ± 0.09	Ash	4.06 ± 0.03
Protein	6.80 ± 0.27	Protein	8.40 ± 0.30
Fat	3.70 ± 0.10	Fat	4.42 ± 0.04
Fiber	38.96 ± 1.64	Fiber	nm *
Carbohydrates	36.21 ± 0.20	Carbohydrates	83.10 ± 0.30
Fructose	nm *	Fructose	2.92 ± 0.05
Glucose	nm *	Glucose	5.36 ± 0.08

* nm—not measured; ** dw—dried weight.

**Table 2 plants-11-01993-t002:** Phytochemical profile of *C. vulgaris*, structure, localization and identification methods.

Territorial Localization	Phytochemical Class	Compound	Part of the Plant	Identification Method	Results					Reference
**UK**	**Phenolic acids**	**chlorogenic acid**	**Shoots**	Chromatography	3.2 g					[118]
	**Tanins**	quercetin 3-*O*-glucoside	Shoots		2.4 g					
		quercetin 3-*O*-galactoside	Shoots		2.4 g					
		quercetin 3-*O*-arabinoside	Shoots		1.3 g					
		(+)-catechin	Shoots/ Roots		1.5 g					
		procyanidin D-1	Shoots/Roots		1.2 g					
		3, 5, 7, 8, 4′-pentahydroxyflavanone 8-*O*-glucoside (callunin)	Shoots		1.1 g					
**Turkey**	**Flavonoid**	kaempferol-3-O-_-d-galactoside	Aerial part	Chromatography	45.7 mg					[96]
**Poland**	**Triterpenoids**	α -Amyrin	Flower and leaves	GC–MS/FID	**Flower cuticular wax**			**Leaf cuticular wax**		[111]
					mg/g wax extract mass ± SD					
					7.8 ± 0.6			8.2 ± 0.7		
		α -Amyrenone			0.5 ± 0.1			3.6 ± 0.4		
		β -Amyrin			6.9 ± 0.3			6.7 ± 0.5		
		Betulin			0.4 ± 0.1			0.6 ± 0.1		
		Cycloartanol			2.8 ± 0.2			4.1 ± 0.3		
		24-Methylenecycloartanol			3.1 ± 0.2			1.2 ± 0.1		
		Erythrodiol			0.6 ± 0.1			2.4 ± 0.2		
		Friedelin			16.8 ± 1.2			1.5 ± 0.1		
		4-Epi-friedelin			2.4 ± 0.2			1.5 ± 0.1		
		Friedelinol			1.0 ± 0.1			1.4 ± 0.1		
		Germanicol			0.2 ± 0.1			3.9 ± 0.3		
		Lupeol			6.9 ± 0.5			7.1 ± 0.6		
		Oleanolic aldehyde			0.6 ± 0.1			1.3 ± 0.1		
		Taraxasterol			5.0 ± 0.1			3.3 ± 0.2		
		Taraxerone			0.8 ± 0.1			1.4 ± 0.1		
		Ursolic aldehyde			3.3 ± 0.2			5.2 ± 0.3		
		Uvaol			2.5 ± 0.2			28.6 ± 2.2		
		Betulinic acid			n.d.			9.4 ± 0.8		
		Oleanolic acid			28.2 ± 1.6			125.1 ± 9.8		
		3-Oxo-olean-12-en-28-oic acid			7.1 ± 0.4			3.1 ± 0.3		
		3-Oxo-ursan-12-en-28-oic acid			1.1 ± 0.1			5.0 ± 0.3		
		Ursolic acid			75.2 ± 4.1			398 ± 25.7		
		Campesterol			1.0 ± 0.1.			0.6 ± 0.1		
		Cholesterol			2.3 ± 0.1			0.8 ± 0.1		
		Sitostanol			3.0 ± 0.1			1.5 ± 0.1		
		Sitosterol			8.1 ± 0.5			13.8 ± 1.1		
		Stigmasterol			0.7 ± 0.1			0.3 ± 0.1		
		Stigmasta-3,5-dien-7-one			4.4 ± 0.3			1.9 ± 0.2		
		Stigmastane-3,6-dione			1.6 ± 0.1			1.4 ± 0.1		
	**Esters**	α -Amyrin			1.0 ± 0.1			1.0 ± 0.1		
		β -Amyrin			0.4 ± 0.1			1.1 ± 0.2		
		Cycloartanol			0.2 ± 0.1			0.7 ± 0.1		
		Lupeol			0.8 ± 0.1			0.9 ± 0.1		
		Taraxasterol			0.5 ± 0.1			0.5 ± 0.1		
		Oleanolic acid			1.0 ± 0.1			0.6 ± 0.1		
		Ursolic acid			2.5 ± 0.2			n.d.		
**Spain**	**Triterpenic acids**	Oleanolic acid	Leaves	HPLC	0.53-82.87 mg/g extract (max value obtained with the highest yield—7.31% and highest ethanol content—15%)					[123]
		Ursolic acid			2.19–141.45 mg/g extract (max value obtained with the highest yield—7.31% and highest ethanol content—15%)					
**Portugal**	**Vitamers**	α-Tocopherol	Flowers	H-NMR and LC-UV-MS	32.50 ± 0.48 mg/100 g					[33]
		β-Tocopherol			0.39 ± 0.01 mg/100 g					
		γ-Tocopherol			1.19 ± 0.06 mg/100 g					
		δ-Tocopherol			0.36 ± 0.01 mg/100 g					
		α-Tocotrienol			nd					
		β-Tocotrienol			nd					
		γ-Tocotrienol			0.54 ± 0.03 mg/100 g					
		δ-Tocotrienol			nd					
	**Tannins**	Quinic acid			27.07 ± 0.11 μg/g					
		Proanthocyanidins trimmer			25.05 ± 0.18 μg/g					
		Proanthocyanidins tetramer			8.87 ± 0.08 μg/g					
	**Flavonoid**	3,5,7,tetrahydroxy-4′-ethoxyflavanone 8-deoxyhexoside			55.14 ± 0.11 μg/g					
		Methoxy myricetin deoxyhexoside			308.52 ± 0.48 μg/g					
		Quercetin deoxyhexoside			101.30 ± 0.34 μg/g					
		Kaempferol deoxyhexoside			2.11 ± 0.03 μg/g					
		Myricetin deoxyhexoside			6.39 ± 0.08 μg/g					
		Quercetin			140 ± 0.09 μg/g					
		Kaempferol			15.36 ± 0.11 μg/g					
		Myricetin			44.62 ± 0.12 μg/g					
	**Tocopherols**	α-Tocopherol	Flowers	HPLC	5.84 ± 0.07 mg/100 g dw					[90]
		β-Tocopherol			0.25 ± 0.0001 mg/100 g dw					
		γ-Tocopherol			0.75 ± 0.03 mg/100 g dw					
		δ-Tocopherol			1.05 ± 0.08 mg/100 g dw					
	**Hydroxycinnamic acids**	5- O-Caffeoylquinic acid		HPLC-DAD- ESI/MS	Ethyl acetate (mg/g extract)	Acetone (mg/g extract)	Methanol (mg/g extract)	Decoction (mg/g extract)	Infusion (mg/g extract)	
					n.d.	0.20 ± 0.02	3.3 ± 0.1	3.1 ± 0.2	5.0 ± 0.2	
		(+)-Catechin			0.52 ± 0.09	n.d.	n.d.	n.d.	n.d.	
	**Flavonoid**	Myricetin-O-hexoside			n.d.	2.12 ± 0.02	2.09 ± 0.01	4.50 ± 0.03	4.34 ± 0.02	
		5-p-Coumaroylquinic acid			n.d.	n.d.	0.10 ± 0.01	0.06 ± 0.01	0.150 ± 0.001	
		Myricetin-3- O-glucoside			1.63 ± 0.07	4.96 ± 0.09	4.30 ± 0.09	5.80 ± 0.01	8.1 ± 0.1	
		Myricetin –O-rhamnoside			1.66 ± 0.09	2.04 ± 0.06	2.81 ± 0.05	4.86 ± 0.06	5.8 ± 0.1	
		Quercetin-3- Oglucoside			0.37 ± 0.03	2.20 ± 0.06	1.68 ± 0.004	1.65 ± 0.001	3.01 ± 0.08	
		Quercetin-O-hexoside			0.355 ± 0.0003	0.8 ± 0.002	0.97 ± 0.07	1.22 ± 0.03	2.05 ± 0.09	
		Isorhamnetin-3- Oglucoside			1.6 ± 0.3	11.25 ± 0.08	2.8 ± 0.1	2.33 ± 0.03	5.48 ± 0.08	
		Quercetin-O-hexosid			0.298 ± 0.005	0.42 ± 0.03	0.350 ± 0.005	0.81 ± 0.001	0.77 ± 0.01	
		Kaempferol-O-rhamnoside			0.36 ± 0.04	n.d.	0.39 ± 0.01	0.83 ± 0.003	0.81 ± 0.01	
		Isorhamnetin-O-rhamnoside			0.428 ± 0.001	0.41 ± 0.01	0.53 ± 0.01	0.92 ± 0.01	1.12 ± 0.03	
**Rusia**	**Hydroxycinnamic acids**	chlorogenic acid	Seeds	HPLC	4.298 ± 0.301 mg/g					[124]
		caffeic acid			0.0036 ± 0.0003 mg/g					
	**Flavonoids**	kaempferol,			0.036 ± 0.030 mg/g					
		quercetin			30.045 ± 2.003 mg/g					
		rutin			3.133 ± 0.219 mg/g					
		myricetin			2.116 ± 0.148 mg/g					
	**Tannins**	epicatechin			1.515 ± 0.106 mg/g					
		catechin			7.679 ± 0.538 mg/g					
**Ukraine**	**Phenols**	Arbutin	Aerial part	HPLC/spectro- photometry	Water extraction (%, n = 5)			Hydroethanolic Extract (%, n = 5)		[102]
					1.25 ± 0.05			0.83 ± 0.04		
		Methylarbutin			0.18 ± 0.03			0.23 ± 0.02		
	**Hydroxy-cinnamic acids**	Chlorogenic		HPLC/spectro- photometry	1.25 ± 0.03			1.74 ± 0.03		
		Caffeic			0.02 ± 0.01			0.03 ± 0.01		
		Ferulic			0.11 ± 0.02			0.12 ± 0.01		
		p-Coumaric			0.03 ± 0.01			0.04 ± 0.01		
	**Flavonoids**	Rutin		HPLC/spectro- photometry	0.65 ± 0.05			1.25 ± 0.05		
		Hyperoside			0.15 ± 0.05			0.20 ± 0.01		
		Quercetin-3-D-glucoside			0.17 ± 0.03			0.29 ± 0.01		
		Luteolin			0			0.05 ± 0.01		
		Apigenin			0			0.04 ± 0.01		
		Kaempferol			0.02 ± 0.01			0.05 ± 0.01		
	**Tannins metabolites**	Gallic acid		HPLC/spectro- photometry	0.07 ± 0.01			0.13 ± 0.01		
		(+)-Gallocatechin			0.21 ± 0.01			0.94 ± 0.02		
		(−)-Epigallocatechin			0.95 ± 0.05			1.36 ± 0.09		
		(+)-Catechin			0.13 ± 0.03			0.21 ± 0.03		
		(−)-Epicatechin			0.09 ± 0.01			0.26 ± 0.02		
		(−)-Catechin gallate			0.11 ± 0.02			0.24 ± 0.01		
		(−)Epicatechin gallate			0.05 ± 0.01			0.07 ± 0.01		

GC–MS/FID—gas chromatography–mass spectrometry with flame-ionization detection; HPLC—high-performance liquid chromatography; n.d.—not detected; dw—dry weight; H-NMR—proton nuclear magnetic resonance; LC–UV–MS—high-performance liquid chromatography coupled with diode array UV detection and mass spectrometry; GC–MS—chromatography–mass spectrometry; HPLC–DAD–ESI/MS—high-performance liquid chromatography coupled with a diode array detector and mass spectrometry using electrospray ionization.

**Table 3 plants-11-01993-t003:** Major pharmacological properties of the phytochemicals present in *C. vulgaris* and their associated disorders.

Pharmacological Properties	Main Compound	Associated Disorder	Model Used	References
Antibacterial	phenolic compounds	Urinary tract pathogens	In vitro	[95]
		Vaginal microbiota		[90,92]
		Human pathogen microorganisms		[28]
				[93]
Antidepressant	quercitin		In vitro	[101]
Anti-anxiety	arbutin, chlorogenic acid, rutin, hyperoside, quercetin-3-D-glucoside, (+)-gallocatechin and (−)-epigallocatechin	Depression	In vivo (rats and mice)	[102]
Neurotropic				
Anxiolytic				
Stress-protective				
Antiviral	oleanolic and ursolic acid	Hepatitis C	In vitro	[123]
Antiproliferative	ursolic acid	Leukemia	In vitro	[139]
Antioxidant	kaempferol-3-*O*-β-D-galactoside	Oxidative- stress-related disorders	In vitro	[91]
	chlorogenic acid			[89]
	phenolic compounds	Human pathogen microorganisms	In vitro	[28,93]
Antihypertensive	chlorogenic acid	Gout	In vivo (rats)	[99]
Analgesic				
Hypouricemic				
Anti-inflammatory	kaempferol-3-*O*--d-galactoside	Gastritis, ulcer	In vivo (mice)	[140]
	phenols and flavonoids	Urinary tract pathogens	In vitro	[95]
	phenols and flavonoids	Alzheimer	In vitro	[98]
	chlorogenic acid	Gout	In vivo (rats)	[99]
Chemoprotective	hyperoside, quercitrin, quercetin, kaempferol	Skin burns and skin cancer	In vivo (mice)	[141]
			In vitro	[97,103]

**Table 4 plants-11-01993-t004:** Physicochemical characteristics of heather honey from different geographical origins.

Geographical Origin	EC (mS/cm)	Free acidity (%)	Pfund (mm)	Moisture (%)	Total Sugars (Brix%)	HMF (mg/kg)	References
**Romania**	0.67 ± 0.17	17.93 ± 3.64	n.m.	20.66 ± 1.02	77.77 ± 7.82	11.23 ± 4.97	[25]
	0.66 ± 0.07	16.90 ± 2.21	n.m.	20.52 ± 0.37	78.47 ± 4.26	10.00 ± 3.02	[193]
**Poland**	0.62	22.10	69.00	19.10	73.70	4.80	[192]
**Turkey**	n.m	4.40 ± 0.03	n.m.	17.98	69.0 ± 0.07	94.66 ± 1.36	[194]
**Portugal**	0.71 ± 0.08	30.89 ± 5.58	n.m.	17.59 ± 0.37	72.16 ± 2.43	7.00 ± 6.68	[172]
	0.66 ± 10.30	14.8 ± 0.76	n.m.	24.20 ± 0.00	73.8 ± 0.10	12.19 ± 0.74	[195]
	0.71 ± 0.00	30.33 ± 1.53	333 ± 0.00	16.47 ± 0.06	82.03 ± 0.06	n.m.	[191]
**Spain**	0.777 ± 0.11	n.m.	116.00 ± 15.00	n.m.	n.m.	n.m.	[196]
**Ireland**	0.603 ± 0.09	n.m.	119 ± 40.85	23.33 ± 1.58	71.46 ± 5.94	n.m.	[186]
**Algeria**	0.750 ± 0.20	34.7 ± 10.20	122.00 ± 7.00	18.40 ± 0.60	n.m.	4.90 ± 0.80	[197]
**Estonia**	0.400 ± 0.20	24.3 ± 8.20	n.m.	18.80 ± 1.40	74.2 ± 2.70	7.70 ± 3.00	[198]

EC—electrical conductivity; HMF—hydroxymethylfurfural; n.m.—not measured; results reported are the average or average ± standard deviation values.

## Data Availability

Not applicable.

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
