# Peer review of "Calluna vulgaris* as a Valuable Source of Bioactive Compounds: Exploring Its Phytochemical Profile, Biological Activities and Apitherapeutic Potential"

_plants, 2022, doi:10.3390/plants11151993_

Round 1
Reviewer 1 Report
Applies to: Calluna vulgaris as a Valuable Source of Bioactive Compounds: Exploring its Phytochemical Profile, Biological Activities and Apitherapeutic Potential
I would like to congratulate the Authors on the meticulous work put into writing this manuscript.
The work presented to me for review is comprehensive, well-thought-out, and worth publishing in Plants.
Nevertheless, I am asking you to correct the following shortcomings:
1 / Figure 1. Please adjust the size, center the caption
2 / units, please standardize throughout the work, or the record: mg/mL or mg mL-1, or mg per mL, chaos reigns now
3 / notation of numbers - please use dots as separators, commas will appear; please round to whole numbers, to decimal or hundredth values - currently it is not order; please remember that to the numbers we "stick" only degrees and %, the remaining units are written with a space
4 / citations - please edit as required by Plants; there are quotations in the sentences from the author's name, but please remember that then you should put a number at the end of the sentence anyway.
5 / If there is a reference to "Table" or "Graph" in the text - please bold
6 / notation of polyphenolic compounds: quercetin 3-0-glucoside -> it is unacceptable to use "0" and not "O" in the notation, please remember that it is an oxygen atom, not zero; please use italics for oxygen atoms
7 / Table 2 - it is very unfortunate - reformat please - why is the last column so wide and nothing fits in the others?
8 / Table 3 - also for editing; there are errors in the record of relationships; please pay attention to details so that the text in the table is uniform and consistent for all tables
9 / Please explain ALL the abbreviations used in the text at the first mention; you can also create a section with abbreviations
10 / When using the abbreviations in the table, please give their explanation below the table each time
11 / Please pay attention to subscripts, e.g. IC50 - it appears once in the index and sometimes not - please standardize
12 / The language of the text is colloquial in some places, please provide the linguistic proofreading certificate
Author Response
Thank you for your suggestions. Please find enclosed the response to each observation.
The manuscript was revised again by all authors and modified according to your suggestions.
Regards
Reviewer 2 Report
Manuscript plants-1827344 compries a critical review on the health benefits and biological activity of Caluna vulgaris. The authors have provided numerous and important literature upon this topic with an original style. In addition, they link the beneficial effects of Caluna with heather honey, as the outcome of honeybees forage in this nectar plant. The collective information presented also for heather honey are of importance for the literature and the readership of the journal.
As far as the technical quality, the review article has been very carefully prepared and is well-structured. Tables need some corrections, while figures are of very good quality. Other corrections that authors should done are indicated in the attached pdf.
Based on these comments, I suggest a minor revision prior to further consideration for publication.

Author Response
Thank you for your observations.
We have revised the manuscript and made the suggested changes.
Regards
